# FedAnchor: Enhancing Federated Semi-Supervised Learning with Label Contrastive Loss

## Abstract

Federated learning (FL) is a distributed learning paradigm that allows devices to collaboratively train a shared global model while keeping the data locally. Due to the nature of FL, it provides access to an astonishing amount of training data for meaningful research and applications. However, the assumption that all of these private data samples include correct and complete annotations is not realistic for real-world applications. Federated Semi-Supervised Learning (FSSL) provides a powerful approach for training models on a large amount of data without requiring all data points to be completely labeled. In this paper, we propose *FedAnchor*, an innovative method that tackles the *label-at-server* FSSL scenario where the server maintains a limited amount of labeled data, while clients' private data remain unlabeled. *FedAnchor* introduces a unique double-head structure, with one *anchor head* attached with a newly designed label contrastive loss based on the cosine similarity to train on labeled anchor data to provide better pseudo-labels for faster convergence and higher performance. Following this approach, we alleviate the confirmation bias and over-fitting easy-to-learn data problems coming from pseudo-labeling based on high-confidence model prediction samples. We conduct extensive experiments on three different datasets and demonstrate our method can outperform the state-of-the-art method by a significant margin, both in terms of convergence rate and model accuracy.

## 1 Introduction

Federated learning (FL) (McMahan et al., 2017) allows edge devices to collaboratively learn a shared global model while keeping their private data locally on the device. There are nearly seven billion connected Internet of Things (IoT) devices and three billion smartphones around the world (Lim et al., 2020), potentially giving access to an astonishing amount of training data and decentralized computing power for meaningful research and applications. Most existing FL works primarily focus on supervised learning where the local private data is fully labeled. However, assuming that the full set of private data samples includes rich annotations is unrealistic for real-world applications (Jeong et al., 2020; Diao et al., 2022; Jin et al., 2020; Yang et al., 2021). Although for some applications of FL, such as keyboard predictions, labeling data requires virtually no additional effort, this is not generally the case.

Acquiring large-scale labeled datasets on the user side can be extremely costly. For example, a large amount of unlabeled data is generated through interactions with smart devices in daily life, such as pictures or physiological indicators measured by wearables. These volumes of data make it impractical to mandate individual users to annotate the data manually. This task can be excessively time-intensive for users, or they may lack the requisite advanced knowledge or expertise for accurate annotation, particularly when the dataset pertains to a specialized domain such as medical data (Yang et al., 2021). The complicated process of annotation results in most user data remaining unlabeled, further leading to the conventional FL pipeline being unable to conduct supervised learning. Recent studies of self-supervised learning in FL attempt to leverage unlabeled user data to learn robust representations (Gao et al., 2022; Rehman et al., 2022; 2023). However, the learned model still requires fine-tuning on labeled data for downstream supervised tasks.

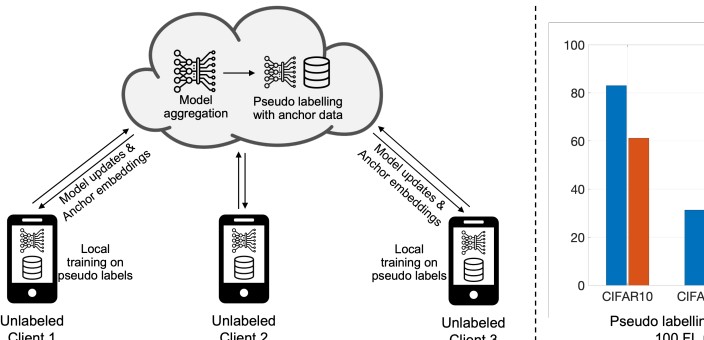

Figure 1: (left) Pipeline of *FedAnchor* with pseudo labeling and anchor data on the server. Anchor embeddings are only transmitted to the clients during downstream communication. (right) Pseudo labeling accuracy with 5000/2500/1000 anchor data on CIFAR10/CIFAR100/SVHN datasets, respectively.

Compared to FL environments, centralized data annotation is more straightforward and precise in data centers. Even in low-resource contexts (e.g., medical data), the task of labeling limited data stored on the server by experts would not demand substantial effort. The integration of semi-supervised learning (SSL) (Chapelle et al., 2009; Yang et al., 2022; Sohn et al., 2020; Berthelot et al., 2019) with FL can potentially leverage limited centralized labeled data to generate pseudo labels for supervised training on unlabeled clients. Existing work (Jeong et al., 2020; Zhang et al., 2021b) attempted to perform SSL by using off-the-shelf methods only, such as FixMatch (Sohn et al., 2020), MixMatch (Berthelot et al., 2019) in FL environments. Although these methods provide certain model convergence guarantees during the FL stage, they cause heavy traffic for data communication due to their per-mini-batch communication protocol. Another recent work, SemiFL (Diao et al., 2022), improves the training procedure by implicitly conducting entropy minimization. This is achieved by constructing hard (one-hot) labels from high-confidence predictions on unlabeled data, which are subsequently used as training targets in a standard cross-entropy loss. However, it is argued that using pseudo-labels based on model predictions might lead to a confirmation bias problem or overfitting to easy-to-learn data samples (Nguyen & Yang, 2023). In addition, the existing methods typically establish a pre-defined threshold, generally set relatively high, to retain only samples with very high confidence. This might lead to slow convergence issues, especially during the beginning of training when there are very limited samples satisfying the threshold.

In this paper, we propose an enhanced federated SSL method, dubbed *FedAnchor* - a newly designed label contrastive loss based on the cosine similarity metric to train on labeled anchor data on the server. In this way, instead of retaining the high-confidence data through solely model prediction in the conventional SSL studies, *FedAnchor for-the-first-time* generates the pseudo labels by comparing the similarities between the model representations of unlabeled data and label anchor data. This provides better quality pseudo-labels as shown in Fig. 1 (right), alleviates the confirmation bias, and reduces over-fitting easy-to-learn data issues. Our contributions are summarized as follows: 1) we propose a *unique* pseudo-labeling method *FedAnchor* for SSL in FL, which leverages the similarities between feature embeddings of unlabeled data and label anchor data; 2) we design a novel label contrastive loss to further improve the quality of pseudo labels; 3) we perform extensive experiments on three different datasets having different sizes of labeled anchor data and show that the proposed methods achieve the state-of-the-art (SOTA) performance and faster convergence rate.

## 2 BACKGROUND

**Federated Learning.** Federated learning (FL) aims to collaboratively learn a global model while keeping private data on the device. Let $G$ be the global model and $L = \{l_k\}_{k=1}^K$ be a set of local models for total K clients. We consider a $C$ class classification problem defined over a compact space $\mathcal{X}$ and a label space $\mathcal{Y} = [C] = \{1, ..., C\}$. Let $f : \mathcal{X} \rightarrow \mathcal{S}$ maps $\mathbf{x}$ to the probability simplex $\mathcal{S}$, $\mathcal{S} = \{\mathbf{z} | \sum_{i=1}^L z_i = 1, z_i \geq 0, \forall i \in [C]\}$ with $f_i$ denoting the probability for the $i$th class. $f$ is parameterized over the hypothesis class $w$, which is the weight of the neural network. $\mathcal{L}(\mathbf{w})$ is the loss function. FL training follows communication rounds. During each round $t$, the server broadcasts the current global model $G^t$ to selected clients. Then, the selected clients train the model

using the local dataset and send the updated local model $L_i^{t+1}$ to the server. The server can then aggregate the local models using techniques, e.g. FedAvg (McMahan et al., 2017).

**Semi-Supervised Learning.** Semi-supervised learning (SSL) is a general problem of learning with partially labeled data, especially when the amount of unlabeled data is much larger than labeled ones (Zhou & Li, 2005; Rasmus et al., 2015). The standard SSL method involves giving unlabeled data pseudo-labels (Lee et al., 2013) and then using these pseudo-labels as hard labels for supervised training. For methods such as MixMatch (Berthelot et al., 2019) and FixMatch (Sohn et al., 2020), the pseudo-label is only retained if the model produces a high-confidence prediction, and then training involves implementation of different strong augmentation methods.

**Federated Semi-Supervised Learning.** Federated semi-supervised learning (FSSL) represents the federated variant of SSL. In this context, the data stored on the client side may or may not be labeled. Given a dataset $\mathcal{D} = \{\mathbf{x}_i, y_i\}_{i=1}^{N}$, $\mathcal{D}$ is split into a labeled set $\mathcal{S} = \{\mathbf{x}_i, y_i\}_{i=1}^{S}$, which will be called *Anchor Data* in our paper; and an unlabeled set $\mathcal{U} = \{\mathbf{x}_i\}_{i=1}^{U}$ as in the standard semi-supervised learning. Taking the off-the-shelf SSL method and directly applying it to FL cannot achieve communication-efficient FL training. For example, techniques such as FixMatch (Sohn et al., 2020) or MixMatch (Berthelot et al., 2019) require each mini-batch to sample from both labeled and unlabeled data samples with a carefully tuned ratio, which is impossible to achieve in the FL settings. FedMatch (Jeong et al., 2020) splits model parameters for labeled servers and unlabeled clients separately. FedRGD (Zhang et al., 2021b) trains and aggregates the model of the labeled server and unlabeled clients in parallel with the group-side re-weighting scheme while replacing the batch normalization (BN) to group normalization (GN) layers. Liu et al. (2021) and Saha et al. (2023) considered the case when there are both labeled and unlabeled data at clients in the real medical settings; imFed-Semi (Jiang et al., 2022) conducts the client training by exploiting class proportion information; FedoSSL (Zhang et al., 2023) proposes a framework tackling the biased training process for heterogeneously distributed and unseen classes; FedSiam (Long et al., 2020) integrates a siamese network into FL, incorporating a momentum update to address the non-IID challenges posed by unlabeled data; FedIL (Yang et al., 2023) employs iterative similarity fusion to maintain consistency between server and client predictions on unlabeled data, and utilizes incremental confidence to construct reliable pseudo-labels. SemiFL (Diao et al., 2022) combines centralized semi-supervised learning method MixMatch (Berthelot et al., 2019) and FixMatch (Sohn et al., 2020) together with an alternate training scheme to achieve the current SOTA performance. FedCon (Long et al., 2021) borrows the idea of using two models from BYOL (Grill et al., 2020) without implementing any contrastive loss, despite the name. Instead, they use a *consistency loss* between two different augmentations to help clients' networks learn the embedding projection. Both SemiFL and FedCon serve as baselines in our paper.

**Latent Representation.** The great success Neural Networks (NN) have achieved since their introduction is to be adjudged to their capability of learning latent representations of the input data that can eventually be used to learn the task they have been designed for. The set containing this latent information is often referred to as *latent space*, which can also be the subject of topological investigation (Zaheer et al., 2017; Hensel et al., 2021). Many theoretical studies have highlighted the importance of such representations as they are explicitly identified in many settings, e.g. the intermediate layers of a ResNet architecture (He et al., 2016), the word embedding space of a language model, or the bottleneck of an Autoencoder (Moor et al., 2020). More recently, the research community focused on investigating the *quality* of the *latent space* as well-performing networks have shown similar learned representations (Li et al., 2015; Morcos et al., 2018; Kornblith et al., 2019; Tsitsulin et al., 2019; Vulić et al., 2020). The most audacious attempt to leverage the structure of different latent spaces is presented in (Moschella et al., 2022), which provided interesting observations regarding the structure of different latent spaces learned by diverse training procedures to achieve *zero-shot model stitching*. Other methods, such as those proposed in (Lee et al., 2018; Zhang et al., 2021a; Li et al., 2021), also used latent representation for tackling problems around noisy labels and data. However, such analyses or methods were never extended to a semi-supervised learning or federated learning context.

## 3 METHODOLOGY: *FedAnchor*

In this section, we present our method *FedAnchor* (Fig. 1) in detail. *FedAnchor* aims to fully utilize the information embedded in the anchor dataset stored on the server to provide better pseudo-labels for unlabeled private client's data to be trained on supervised tasks. We propose a novel label

contrastive loss combined with cosine similarity metrics to extract the anchor information in the latent space. The section is organized as below: we detail the new label contrastive loss in Section 3.1; we then explain in Section 3.2 how to obtain the pseudo label in our methods; then it is followed by the algorithm of local training on the client side in Section 3.3 and the server training using anchor data in Section 3.4. The pseudo-code of the *FedAnchor* is summarized in Algorithm 1.

## 3.1 LABEL CONTRASTIVE LOSS

One of the main novelties of our method lies in introducing new label contraction loss on the latent space. As mentioned in the background section (Section 2), the latent space contains important representations that can be identified and utilized in many settings. In our case, the latent space can be the output of any pre-defined layers in the neural network.

We propose to have a double-head structure for the model, with one head (*classification head*) consisting of the original classification layers and the other (*anchor head*) consisting of a projection layer to the latent space to reduce the previous embedding dimension. The *anchor head* is crucially designed for the new label contrastive loss that we propose below. This structure is model agnostic and can be implemented in all the deep learning model architectures used for a classification task.

We define the $x^m$ to be the unlabeled data samples on clients $m$; $(x^{anchor}, y^{anchor})$ be the labeled anchor data; $z^m$ be the output of the latent space/anchor head for the unlabeled client data, and $z^{anchor}$ be the output of the latent space/anchor head for the anchor data. Hence, $z$ represents the latent representation of any given data. Also, we let $\hat{y}$ denote the pseudo-label for unlabeled data.

The objective is to map the $z^{anchor}$ of the same label to the same localized region in the latent space while forcing data with different labels to be far from each other. We let $s_{i,j}$ be the similarity function between data sample $x_i$ and $x_j$ in the latent space $(z_i, z_j)$. Cosine similarity is used in our case: $s_{z_i, z_j} = \frac{z_i^T z_j}{\|z_i\|\|z_j\|}$.

Then, we propose our new label contrastive loss as in eq. 1 and 2. The label contrastive loss is defined on a batch of anchor data samples. Given a batch of anchor data, we calculate the $l(c)$ for each label class, then sum up all label classes to obtain the final label contrastive loss value.

$$l(c) = -\log \frac{\sum_{y(i), y(j)=c} \exp(s_{z_i, z_j}/\tau)}{\sum_{y(i) \neq y(j)} \exp(s_{z_i, z_j}/\tau)} \tag{1}$$

$$\mathcal{L}_c = \frac{1}{C} \sum_{c=1}^{C} l(c) \tag{2}$$

where $\tau$ is the temperature hyper-parameter that can be tuned. The label conservative loss aims to maximize the similarities between samples with the same labels while minimizing the similarities between samples with different labels. The proposed loss will eventually force data samples with the same label to be projected in the same localized region of the latent space while forcing regions referring to different labels to be far from each other.

## 3.2 PSEUDO-LABELING USING ANCHOR HEAD

We present our pseudo-labeling method by describing the steps composing one communication round, which will be iterated during the FL process. At the communication round $t$, the server selects a subset of clients to participate with participation ratio $r$ in the current federated round. The server will then broadcast the model parameters ($w_t$) and the latent anchor representations ($\{z_i^{anchor}\}_{i=1}^{S}$) to the selected clients. The anchor latent representations are the output of the *anchor head*, which is used for generating the hard pseudo-label during training.

After receiving the current model weights and anchor latent representations, each client $m$ first compute the latent representation of each local data ($\{z_i^m\}_i$). Then, it compares $z_i^m$ to each anchor latent representation ($\{z_i^{anchor}\}_{i=1}^{S}$) to obtain the pseudo-label by computing the cosine similarity between $z_i^m$ and each anchor latent representation. The scores obtained from this comparison are averaged by label, i.e. cosine similarities relative to anchor latent representation from anchor samples with the same label are averaged. The pseudo-label is the label that provides the maximum score.

Let $Z_c^{anchor} = \{z_i^{anchor}, y_i^{anchor} = c\}$ be the set of anchor latent representation with label $c$. Then, the average cosine similarities between $z_i^m$ and each $Z_c$ can be computed. Let $s^{avg}$ be the average similarities of a data input compared with anchor data. In particular, $s_c^{avg}(z_i^m)$ be the average cosine similarities between unlabeled data latent representation $z_i^m$ compared with each anchor data with label class $c$. The value of $s_c^{avg}(z_i^m)$ can be calculated as:

$$s_c^{avg}(z_i^m) = \frac{1}{|Z_c^{anchor}|} \sum_{z_k \in Z_c^{anchor}} s(z_i^m, z_k), \tag{3}$$

Subsequently, to obtain the pseudo-label $\hat{y}_i^m$ for each local data, we need to find the label class that produces the maximum $s_c^{avg}(z_i^m)$:

$$\hat{y}_i^m = \underset{c \in [C]}{\arg\max}(s_c^{avg}(z_i^m)). \tag{4}$$

### 3.3 LOCAL TRAINING

After obtaining the pseudo-labels $(\hat{y}^m)$ for the local data, each local client will then construct a high-confidence dataset $\mathcal{D}_m^{fix}$, which is called the *fix dataset* inspired by FixMatch (Sohn et al., 2020) and SemiFL (Diao et al., 2022). The *fix dataset* is defined to be the set of data samples with the similarity scores above the preset threshold $t$:

$$\mathcal{D}_m^{fix} = \{(x_i^m, \hat{y}_i^m) \ \texttt{s.t.} \ \max(s_c^{avg}(z_i^m)) > t\}. \tag{5}$$

The current local training will be stopped if some clients have an empty *fix dataset*. Otherwise, we then will sample with replacement to construct a *mix* inspired by MixMatch (Berthelot et al., 2019) as below:

$$\mathcal{D}_m^{mix} = \texttt{Sample}|\mathcal{D}_m^{fix}| \ \texttt{with replacement from} \ \{(x_i^m, \hat{y}_i^m)\} \tag{6}$$

where $|\mathcal{D}_m^{fix}|$ represents the size *fix dataset*, and in this case $|\mathcal{D}_m^{fix}| = |\mathcal{D}_m^{mix}|$.

During local training with a nonempty fix dataset, the loss function $\mathcal{L}$ consists of two parts: $\mathcal{L}_{fix}$ and $\mathcal{L}_{mix}$. $\mathcal{L}_{fix}$ is calculated as in standard supervised training with mini-batch sampled from the *fix dataset*, but attaching strong augmentation $A(\cdot)$ on each data input:

$$\mathcal{L}_{fix} = l(f(A(x_b^{fix}), \hat{y}_b^{fix}). \tag{7}$$

where $l$ is the loss function, such as cross-entropy loss for classification tasks.

In addition, the *mix loss* $\mathcal{L}_{mix}$ is computed following the Mixup method. Client $m$ constructs a mixup data sample from one fix data $x^{fix}$ and one *mix dataset* $x^{mix}$ by:

$$\lambda_{mix} \sim Beta(a, a), \ x_{mix} = \lambda_{mix} x^{fix} + (1 - \lambda_{mix}) x^{mix}, \tag{8}$$

where $Beta(a, a)$ represents the beta distribution and $a$ is a *mixup hyperparameter*, and the *mix loss* is calculated as:

$$\mathcal{L}_{mix} = \lambda_{mix} \cdot l(f(\alpha(x_{mix}, \hat{y}^{fix})) + (1 - \lambda_{mix}) \cdot l(f(\alpha(x_{mix}, \hat{y}^{mix})), \tag{9}$$

where $\alpha(\cdot)$ represents weak augmentation of data samples. A single local epoch of a client $m$ corresponds to applying as many local SGD steps on the combined loss as the number of batches it has in $\mathcal{D}_m^{mix}$:

$$\mathcal{L}_{combine} = \mathcal{L}_{fix} + b\mathcal{L}_{mix}, \tag{10}$$

where $b$ can be a linear combination coefficient and set to be 1 as default.

Finally, after operating for $E$ local epoch, client $m$ returns the updated model parameters to the central server to finish the local training.

### 3.4 SERVER TRAINING

We aim to use the labeled anchor data on the server as training data for supervised and label contrastive loss to leverage the information they carry fully. Training at the server on the labeled data is not novel as SemiFL (Diao et al., 2022) performs this dubbing *alternate training* procedure. However, by comparison, *FedAnchor* trains on the anchor data at the server in two epochs: one for the supervised classification loss and one for the label contrastive loss. Let $\mathcal{L}_s$ be the supervised training loss, such as the standard cross-entropy loss for the classification task, and $\mathcal{L}_c$ be the label contrastive loss described in Section 3.1. Therefore, the server will train for one epoch on the *classification head* by minimizing the loss $\mathcal{L}_s$ and for one epoch on the *anchor head* by minimizing the loss $\mathcal{L}_c$.

**Algorithm 1** *FedAnchor*: Let $N$ be the total number of clients, with total $n$ data samples; $(x^m)$ be the unlabeled data in client $m$; $(x^{anchor}, y^{anchor})$ be the labeled anchor data in the server; $z^m$ be the output of the latent space for the unlabeled client data, and $z^{anchor}$ be the output of the latent space for the anchor data; we let $\hat{y}$ be the pseudo-label for unlabeled data; $E$ be the number of local epochs; $r$ be the participation ratio of clients in each round; $w_t$ be the aggregated weights at round $t$; $C$ be the total number of labels; $Z^{anchor}$ be the set of anchor latent representation; $\eta_{clt}$ and $\eta_{ser}$ be the client and server learning rate respectively.

**procedure** SERVER EXECUTES
    Initialize the model weight $w_0$
    **for** t = 1,...,T **do**
        $M_t \leftarrow$ random select $(rN)$ number of clients
        compute $Z^{anchor}$
        **for** each $m \in M_t$ **do**
            $w_{t+1}^m \leftarrow$ ClientUpdate$(k, w_t, x_m, Z^{anchor})$
    $w_{t+1} \leftarrow \sum_{m \in M_t} \frac{n_m}{\sum n_m} w_{t+1}^m$
    $w_{t+1} \leftarrow$ ServerUpdate$(w_{t+1})$
**procedure** CLIENTUPDATE$(k, w_t, x_m, Z_a)$
    $\hat{y}_m \leftarrow$ PseudoLabel$(x_m, Z_a)$
    create fix dataset: $\mathcal{D}_m^{fix} = \{(x_{m,i}, \hat{y}_{m,i})$ s.t. $\max(sa(z_{m,i})_c) > t\}$ (eq. 5)
    **if** $\mathcal{D}_m^{fix} \neq \emptyset$ **then**
        create mix dataset: $\mathcal{D}_m^{mix} =$ Sample $|\mathcal{D}_m^{fix}|$ with replacement from $\{(x_{m,i}, \hat{y}_{m,i})\}$ (eq. 6)
    **else**
        terminate ClientUpdate
    **for** local epoch $e = 1, ..., E$ **do**
        $\mathcal{L}_{fix} = l(f(A(x_b^{fix}), \hat{y}_b^{fix})$ (eq. 7)
        $\lambda_{mix} \sim Beta(a, a)$, $x_{mix} = \lambda_{mix} x^{fix} + (1 - \lambda_{mix}) x^{mix}$ (eq. 8)
        $\mathcal{L}_{mix} = \lambda_{mix} \cdot l(f(\alpha(x_{mix}, \hat{y}^{fix})) + (1 - \lambda_{mix}) \cdot l(f(\alpha(x_{mix}, \hat{y}^{mix}))$ (eq. 9)
        $\mathcal{L}_{combine} = \mathcal{L}_{fix} + b\mathcal{L}_{mix}$ (eq. 10)
        $w_{t+1}^m \leftarrow w_t - \eta_{clt} \nabla \mathcal{L}_{combine}$
**procedure** PSEUDOLABEL$(x_m, Z_a)$
    create label specific anchor projected set: $Z_c^{anchor} = \{z_i^{anchor}, y_i^{anchor} = c\}, \forall c \in [C]$
    $s_c^{avg}(z_i^m) = \frac{1}{|Z_c^{anchor}|} \sum_{z_k \in Z_c^{anchor}} s(z_i^m, z_k)$ (eq. 3)
    $\hat{y}_m^m = \arg\max_{c \in [C]} (s_c^{avg}(z_i^m))$ (eq. 4)
**procedure** SERVERUPDATE$(w_{t+1})$
    compute supervised loss $\mathcal{L}_s = l(f(A(x_a), y_a)$
    $w_{t+1} \leftarrow w_{t+1} - \eta_{se} \nabla \mathcal{L}_s$
    $l(c) = -\log \frac{\sum_{y(i), y(j)=c} \exp(s_{z_i, z_j}/\tau)}{\sum_{y(i) \neq y(j)} \exp(s_{z_i, z_j}/\tau)}$ (eq. 1)
    $\mathcal{L}_c = \frac{1}{C} \sum_{c=1}^{C} l(c)$ (eq. 2)
    $w_{t+1} \leftarrow w_{t+1} - \eta_{se} \nabla \mathcal{L}_c$

## 3.5 POSSIBLE ADDITIONS

There are potentially some extensions that can be added to the above-described *FedAnchor* method. One addition can be made to the supervised training on the server. Instead of only using strong augmentation, we can implement the same idea of *fix* and *mix dataset* with loss function as explained eq. 10. The *mixup* method trains a neural network on convex combinations of pairs of examples and their labels with coefficients generated by the beta distribution, which can improve the robustness of the model and utilize the limited labeled data. In this case, the *fix dataset* will be the full anchor dataset, and we sample from the full anchor dataset with replacement to form the *mix dataset*.

Additionally, at the pseudo-labeling stage (Section 3.2), instead of feeding the raw and original unlabeled training data, we can borrow the idea of consistency regularization and pseudo-label ensembles techniques (Bachman et al., 2014; Sajjadi et al., 2016) to weakly augment the unlabeled training data $(\alpha(x_i^m))$ for a few times and then take the ensembles to generate more robust pseudo-labels.

Table 1: Comparison of *FedAnchor* with the SOTA methods and fully supervised baseline. *FedAnchor (mix)* represents the experiments conducted using *FedAnchor*, but the supervised training on the server with anchor data is replaced with the *mixup* method explained in Section 3.5. Supervised models are trained with standard FL procedure without pre-training on anchor data.

| Datasets | | CIFAR10 | | | CIFAR100 | | SVHN | |
|---|---|---|---|---|---|---|---|---|
| Number of anchor data | | 250 | 500 | 5000 | 2500 | 10000 | 250 | 1000 |
| IID ($Dir = 1000$) | Supervised | 89.45±0.47 | 89.73±0.09 | 89.07±0.22 | 61.84±0.17 | 60.56±0.13 | 95.38±0.03 | 94.87±0.53 |
| | FedCon | 34.94±0.43 | 50.81±3.21 | 74.95±1.26 | 32.84±0.40 | 50.05±0.34 | 54.83±2.77 | 83.92±1.03 |
| | FedAvg+FixMatch | 33.98±1.77 | 49.18±2.33 | 75.42±0.73 | 32.31±0.83 | 49.15±0.57 | 43.61±0.64 | 81.65±1.83 |
| | SemiFL | 77.82±0.49 | 81.19±0.35 | 75.46±0.19 | 48.20±0.63 | 63.68±0.16 | 91.55±0.77 | 90.11±1.17 |
| | FedAnchor | 80.36±0.18 | **85.94±0.11** | 83.52±0.41 | 50.79±0.27 | 62.02±0.24 | **91.74±0.41** | **92.77±0.11** |
| | FedAnchor (*mix*) | **82.82±0.21** | 85.87±0.25 | **84.43±0.36** | **51.34±0.07** | **63.99±0.39** | 87.46±0.63 | 92.71±0.54 |
| Non-IID ($Dir = 0.1$) | Supervised | 75.42±5.64 | 77.96±2.55 | 77.99±1.24 | 50.87±1.64 | 48.68±5.32 | 87.48±4.78 | 89.55±0.14 |
| | FedCon | 38.46±0.42 | 51.57±1.34 | 76.38±1.36 | 32.00±0.46 | 48.61±0.56 | 50.86±1.50 | 83.40±1.89 |
| | FedAvg+FixMatch | 39.10±0.17 | 49.92±2.49 | 73.17±1.33 | 34.43±0.87 | 49.53±0.56 | 47.09±1.31 | 76.83±3.26 |
| | SemiFL | 58.82±0.72 | 68.96±0.98 | 72.12±0.35 | 42.41±0.47 | 59.72±0.31 | 68.97±13.24 | 87.21±1.66 |
| | FedAnchor | 60.19±0.32 | 72.75±0.63 | 81.37±0.31 | 43.50±0.13 | 59.96±0.40 | **77.42±0.55** | **90.20±0.56** |
| | FedAnchor (*mix*) | **62.94±0.52** | **73.02±0.31** | **83.59±0.46** | **46.39±0.36** | **61.01±0.06** | 60.30±5.34 | 87.28±0.08 |

## 4 EXPERIMENTS

### 4.1 EXPERIMENTAL SETUP

**Federated datasets**. We conduct experiments on CIFAR-10/100 (Krizhevsky et al., 2009) and SVHN (Netzer et al., 2011) datasets. The datasets are randomly split into labeled anchor data and unlabeled clients' data. To make a fair comparison, we set the number of labeled anchor data samples for CIFAR-10/100 and SVHN datasets to be {250, 500, 5000}, {2500, 10000} and {250, 1000} respectively, according to popular SSL setups (Sohn et al., 2020; Berthelot et al., 2019). To simulate a realistic *cross-device* FL environment using the rest of the data, we generate IID/non-IID versions of datasets based on actual class labels using a Dirichlet coefficient $Dir$ (Reddi et al., 2021), where a lower value indicates greater heterogeneity. As a result, the datasets are randomly partitioned into 100 shards with $Dir = 1000/0.1$ for both IID/non-IID settings.

**Training hyper-parameters**. We select ResNet-18 as the backbone model for all datasets, with the *anchor head* set to be a linear layer with 128 dimensions. During each FL round, 10 clients are randomly selected to participate in the training for 5 local epochs. The FL training lasts for 500 rounds. We increase the number of FL rounds to 800 only for the SVHN 250 anchor case as it seems that 500 is insufficient for the baseline to converge. More details regarding hyperparameters and baseline implementations can be found in Appendix A.1.

### 4.2 COMPARISON WITH SOTA

Table 1 shows the performance of our two *FedAnchor*'s methods along with the current SOTA SemiFL (Diao et al., 2022), FedCon (Long et al., 2021) and the supervised baseline. First, *FedAnchor* achieves new SOTA in all settings and datasets (with some even outperforming the supervised baseline). Specifically, training with a minimal number of anchor data samples (e.g. 250) can yield satisfactory performance in IID FL settings on relatively simple datasets (CIFAR10 and SVHN). Increasing the anchor data can drastically boost performance in the more challenging but realistic non-IID setting. In undertaking more complex tasks like the CIFAR100 dataset, *FedAnchor* necessitates a larger amount of anchor data to achieve elevated accuracy, as has been done in many previous literature. Indeed, this condition is not difficult to meet as numerous labeled data, suitable to be used as anchors, are stored in centralized data centers.

In addition, using the *mixup* method in the server training process obtains slightly enhanced performance in most cases. This improvement is primarily attributable to the ample use of data, which is advantageous for training a more robust model (Berthelot et al., 2019). Utilizing *mixup* method locally on the client-side training can also boost the performance. This can also explain why some *FedAnchor* results can be higher than the fully supervised performance, where no *mixup* is im-

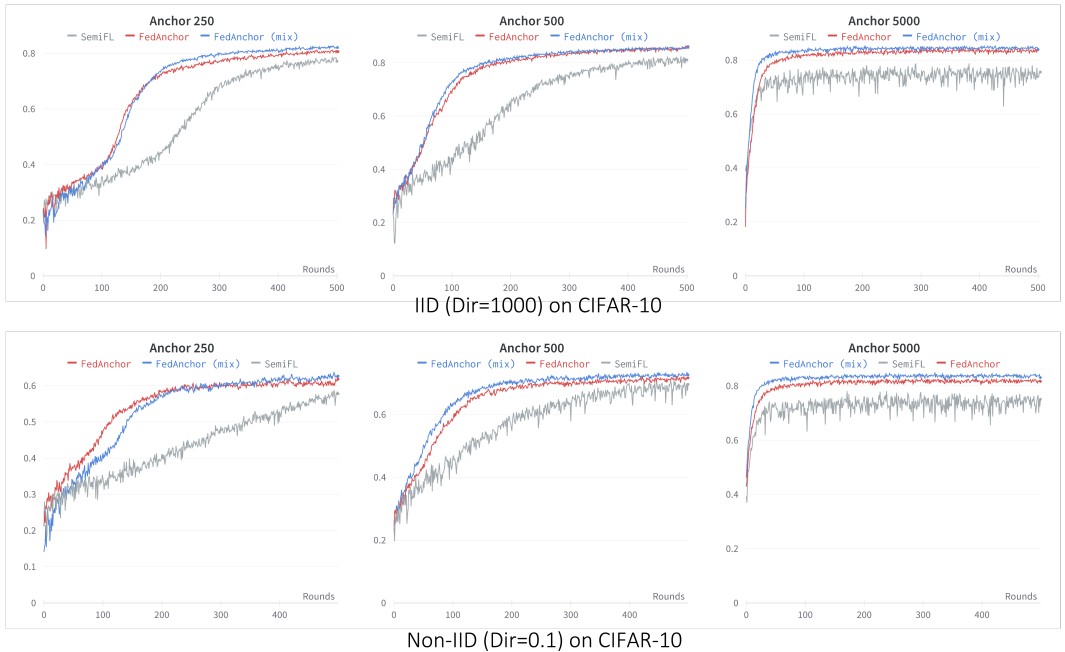

Figure 2: Performance on CIFAR10 test set with both IID and non-IID cases across different numbers of anchor data setups, compared with the baseline.

plemented. Another reason that *FedAnchor* might achieve higher performance can result from the alternate server training process. FL training with non-IID clients can be slow in convergence as shown in many previous research (Li et al., 2020; Karimireddy et al., 2020), even with supervised learning. However, with centralized anchor training, the divergence from the non-IID local datasets might be alleviated every round, resulting in better performance.

Compared to the baseline SemiFL and FedCon, our methods provide more stable performance with less standard deviation. Indeed, this proves beneficial when deploying this method in real-world applications or industrial contexts. The experimental results for SemiFL under SVHN non-IID with 250 anchor data is extremely volatile and slow to converge. It largely depends on the random process, yielding much unstable accuracy. FedCon consistently under-performs in comparison to SemiFL.

Furthermore, the rate of model convergence emerges as a critical concern in FL studies. A slow convergence rate will result in high communication costs and significantly increase the training latency and wall-clock time. Fig. 2 shows the model converging behavior of our proposed methods and the SemiFL baseline. *FedAnchor* makes the faster convergence in all settings. For instance, in the CIFAR10 250 anchor data IID scenario, *FedAnchor* can obtain $60\%$ accuracy within $180$ rounds, while SemiFL requires nearly $300$ rounds to achieve the same performance. The convergence difference is also significant in other datasets.

Additional plots for CIFAR100 and SVHN can also be found in Appendix A.2 Fig. 4 and Fig. 5 respectively. As mentioned in the experimental setup section, we increased the number of training rounds for the SVHN 250 anchor case to $800$ because $500$ rounds are not enough for the baseline to converge. For the SVHN 250 anchor case, *FedAnchor* can achieve $87.4\%$ at round number $200$, while the SemiFL baseline still oscillates around $20\%$ test accuracy level. We can see from Fig. 5 that the test accuracy of SemiFL oscillates around a similar level in the first $100$ rounds.

Furthermore, *FedAnchor* only requires a minor amount of anchor data on the server to obtain acceptable performance. The major overhead in the pseudo-labeling procedure is the computation of the $n * m$ cosine similarities, where $n$ is the number of samples on the client and $m$ those on the server. Cosine similarity is well-known for being low complexity and highly optimizable (Novotný, 2018). Such $n * m$ operations are independent and completely parallelizable. Also, the dimension of

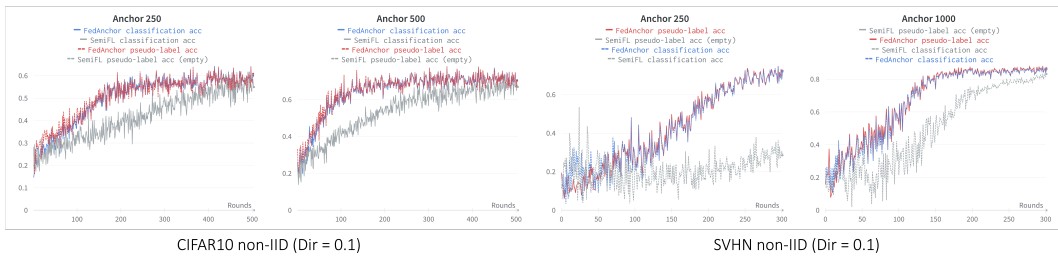

Figure 3: Comparison of pseudo-label accuracy with the *classification head* (used for pseudo-labeling in the baseline) for different datasets/settings.

such vectors is a constant, 128 in the experiments, which is very small. As a result, it is feasible to scale up. In addition, since the server is sharing anchor embeddings with selected clients, there will be downstream communication overhead. We have computed the overhead as shown in Appendix A.3, showing that the overhead is minimal in all cases.

### 4.3 PSEUDO LABELING QUALITY

The quality of pseudo-labels can largely determine the convergence rate of the training and its resulting performance. As unlabeled client training data is trained with generated hard pseudo-labels, with higher pseudo-label accuracy, the model can extract more useful information from the unlabeled client data, hence convergence to better performance with a faster convergence rate. We measure the quality of pseudo-labeling in two different aspects. We compare our pseudo-label accuracy with one of the baseline (SemiFL) pseudo-label accuracy in Fig. 3 and earlier in Fig. 1 (right). Grey line in Fig. 3 shows the pseudo-label accuracy of the baseline while the red line shows the pseudo-label accuracy of *FedAnchor*. It is clear that *FedAnchor* produces significantly higher pseudo-label accuracy than the baseline. Hence, *FedAnchor* can achieve higher performance with a faster convergence rate. Fig. 1 (right) shows the pseudo-label accuracy at the round number 100, demonstrating that the *FedAnchor* can produce higher pseudo-label accuracy with a big margin.

### 4.4 ABLATION STUDIES

We conduct ablation studies on the naive implementation of FixMatch (Sohn et al., 2020), the centralized SOTA method, under the FL setups, as shown in Table 1, to show the effect of the label contrastive loss and the proposed pseudo-labeling. As mentioned in Section 2, the original centralized setting of FixMatch requires sampling both labeled and unlabeled data per mini-batch. Therefore, implementing FedAvg+FixMatch combines FixMatch locally on the client side and alternative training using anchor data on the server side. We can see from the Table that our method significantly outperforms this baseline.

## 5 CONCLUSION

In this paper, we propose *FedAnchor*, which is a FSSL method enhanced by a newly designed label contrastive loss based on the cosine similarity to train on labeled anchor data on the server. In this way, instead of retaining the high-confidence data through solely model prediction in the conventional SSL studies, *FedAnchor* generates the pseudo labels by comparing the similarities between the model representations of unlabeled data and label anchor data. This provides better quality pseudo-labels, alleviates the confirmation bias, and reduces over-fitting easy-to-learn data issues. We perform extensive experiments on three different datasets with different sizes of labeled anchor data on the server and show that the proposed methods achieve state-of-the-art performance and a faster convergence rate. As for future direction, we are experimenting with a fixed threshold during the pseudo-labeling using the anchor data stage. More advanced adaptively and dynamic thresholding techniques can be implemented to potentially further improve the performance and convergence rate.

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

# A APPENDIX

## A.1 TRAINING HYPER-PARAMETERS

Before FL, the model is pre-trained on anchor data on the server for 5 epochs with SGD to speed up the training process with a learning rate of $0.05$. We use a learning rate of $0.03$, a weight decay of $5e-4$, and an SGD momentum of $0.9$ for both local training on the client side and anchor training on the server side. The threshold is set to $0.6$ for our method and $0.95$ for the SemiFL baseline, as indicated in the paper. We conduct three random experiments for all the datasets with different seeds, and the performance is reported on the centralized test set. The hyperparameter for the beta distribution of *mixup* (eq.8) is $a = 0.75$, and the linear coefficient combining *fix loss* and *mix loss* $b$ (eq. 10) is 1 for all experiments. We implement RandAugment (Cubuk et al., 2020) as a robust augmentation method. Our implementation of FedCon (Long et al., 2021) is based on the original GitHub repository ([1]) from which we extracted both the client and server training pipeline and put them in our codebase. We replaced the original backbone with ours to perform a fair comparison.

We have also included the federated supervised training as one of the baselines. The supervised implementation follows the standard experimental protocol as per previous papers (Reddi et al., 2021; Qiu et al., 2022; Horvath et al., 2021), where no strong augmentation methods are implemented.

## A.2 PERFORMANCE W.R.T. FL ROUNDS

In this section, we provide some additional plots for both CIFAR100 and SVHN.

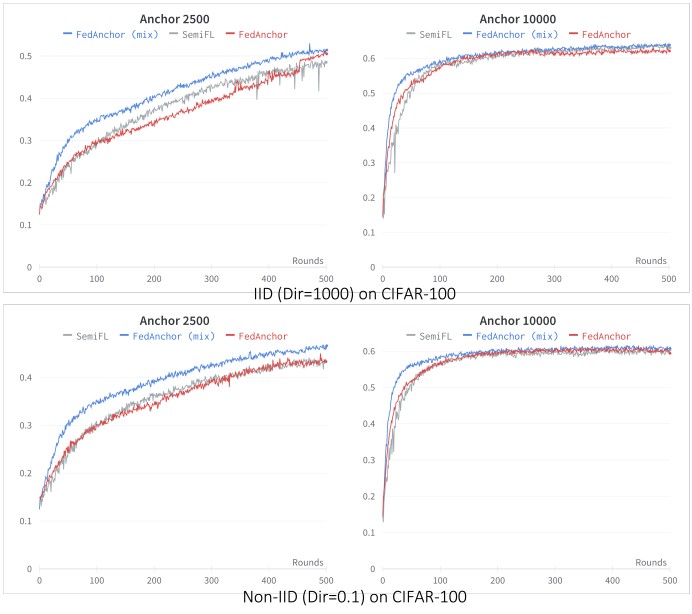

Figure 4: Performance on CIFAR100 test set with both IID and non-IID cases across different numbers of anchor data setups, compared with the baseline.

## A.3 COMMUNICATION OVERHEAD

Table 2 shows the communication overhead of *FedAnchor* compared to standard supervised FL training. The overhead is only for downstream communication when extra anchor embeddings must be sent from the central server to the selected clients. The overhead is calculated as the percentage compared with the supervised FL training when only the model parameters are shared for downstream communication.

---

[1] urlhttps://github.com/zewei-long/fedcon-pytorch

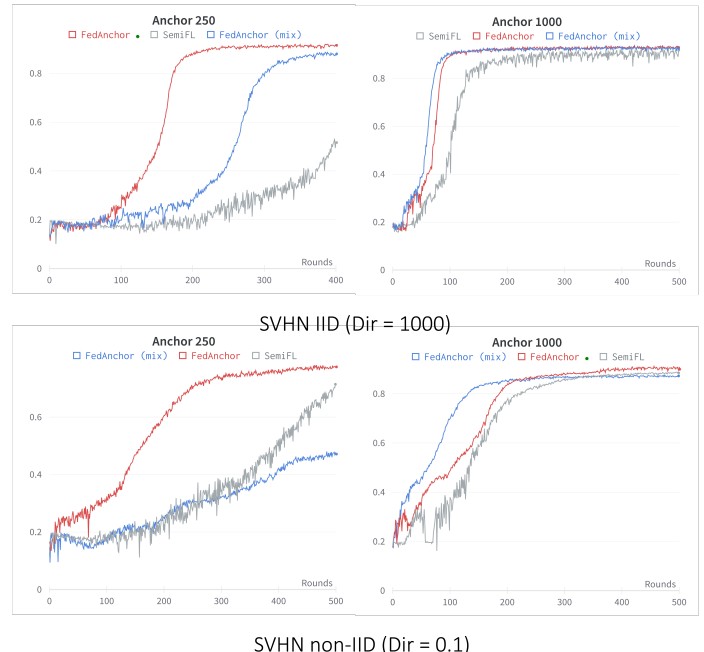

SVHN IID (Dir = 1000)

SVHN non-IID (Dir = 0.1)

Figure 5: Performance on SVHN test set with both IID and non-IID cases across different numbers of anchor data setups, compared with the baseline.

Table 2: Downstream communication overhead of *FedAnchor* compared with supervised FL training. We computed the overhead as the percentage of transmitted parameters that compose the anchors' embeddings over the number of the model's parameters transmitted by *FedAvg*: $Overhead = \frac{100}{\dim(\bar{w})} \sum_{i=1}^{S} \dim(z_i^{anchor})$, where $\bar{w}$ is the model transmitted by *FedAvg*. The upstream communication is the same as *FedAvg*, so there is no overhead.

| Datasets | Anchor Size | Overhead |
|---|---|---|
| CIFAR10 | 250 | 0.29% |
| | 500 | 0.57% |
| | 5000 | 5.73% |
| CIFAR100 | 2500 | 2.85% |
| | 10000 | 11.41% |
| SVHN | 250 | 0.29% |
| | 1000 | 1.15% |

### A.4 FEDCON PLOT

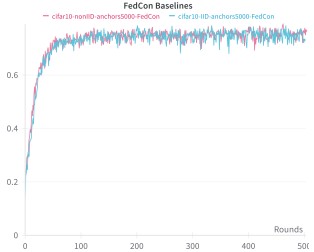

Figure 6: Performance on CIFAR10 test set with both IID and non-IID anchor size 5000 case for FedCon.