# OpenReview forum: "FedAnchor: Enhancing Federated Semi-Supervised Learning with Label Contrastive Loss"
_ICLR.cc/2024/Conference — Submitted to ICLR 2024_

### Official Review · Reviewer_6dEL · 2023-10-25

**Soundness:** 3 good
**Presentation:** 1 poor
**Contribution:** 2 fair
**Rating:** 3
**Confidence:** 4

**Summary:**

This paper considers semi-supervised learning in the federated learning framework, focusing on scenarios where only the server possesses limited labeled data, while clients hold unlabeled data.
The proposed approach introduces a label contrastive loss in addition to the standard cross-entropy loss for server updates.
The server embeddings are then communicated to unlabeled clients to generate pseudo-labels for client supervised learning.
Empirical evaluations are conducted under both IID and non-IID settings to compare the performance of the proposed method against existing approaches.

**Strengths:**

The paper's presentation of the proposed method is straightforward and easy to follow.

**Weaknesses:**

- Method:

    - In the introduction (P2), the authors highlight the potential drawback of some existing works, citing "heavy traffic for data communication". However, the proposed method still necessitates the communication of embeddings of all server data to all active clients in each communication round. While the cost is lower compared to the aforementioned existing methods, the communication overhead remains considerable.

    - The paper claims novelty in the use of the label contrastive loss. However, the experiment does not explicitly demonstrate how this loss contributes to performance improvement. An ablation study could be beneficial to gauge the impact on method performance if the label contrastive loss were omitted or replaced by alternative loss functions. The current description underscores the novelty of the proposed method.


    - Too many hyper-parameters. In the local training (Section 3.3) step, various components contributing to the combined loss function. Parameters such as the threshold value, hyper-parameters for the Beta distribution, strong augmentation method, and size of the mixed dataset, among others, are introduced. However, the paper does not clarify how these hyper-parameters are selected or their potential effects on model performance.

- Experiment:
    - The experiment results in Table 1 for CIFAR100 with 10000 anchor data are puzzling, as the performance of the semi-supervised learning method markedly surpasses that of the supervised learning method in the IID case. This anomaly warrants clarification.


    - The description for Figure 3 is unclear. It is not apparent what the lines represent. Additionally, it is ambiguous whether pseudo-label accuracy and classification accuracy are averaged over all datasets or pertain to a single client. Furthermore, the legends for the two datasets are inconsistent.


- Other issues:
    - There is a notable absence of related work such as [1] -- [4]. These works should be compared to or at least discussed in the related work.

    - Including a descriptive table to highlight the advantages of the proposed method over existing approaches would enhance the paper.


    - The paper needs to be proofread more carefully. Instances of notation inconsistency, key notation typos, and repeated reference entries should be addressed. For instance:

        - On P3, the paragraph discussing Federated Learning uses two separate notations, $G^t$ and $L_i^{t}$, to represent global and local models, respectively. These notations are inconsistent with the rest of the manuscript where the weight 'w' is used to denote different models.

        - The notation for the unlabeled set in the paragraph on Federated Semi-Supervised Learning (P3) is clearly incorrect.

        - The format of the bibliography entries is inconsistent, with some entries including URLs while others do not.

        - There are repeated reference entries for FedMatch (Jeong et al., 2020).


References:

[1] FedCon: A contrastive framework for federated semi-supervised learning

[2] Federated semi-supervised medical image classification via inter-client relation matching

[3] Dynamic bank learning for semi-supervised federated image diagnosis with class imbalance

[4] Towards unbiased training in federated open-world semi-supervised learning

[5] Rethinking semi-supervised federated learning: How to co-train fully-labeled and fully-unlabeled client imaging data

**Questions:**

Please refer to the questions in the previous section.

---

> ### Author Response · Authors · 2023-11-17
>
> We sincerely thank the reviewer for their helpful comments and remarks. We have now addressed all of the concerns below.
>
> - Concern about the communication load: The communication load is minor (please refer to A.3, Table 2 in our revised manuscript).
> - Concern about too many hyper-parameters: The hyper-parameters of beta, augmentation method, and size of the mixed dataset remain the same as the existing papers (such as FixMatch, MixMatch, and SemiFL). As for the threshold value, it could be dataset-dependent. We selected the value 0.6 on CIFAR 10 and kept the same for other datasets to show the generalization of our method.
> - Concern about the supervised baseline: For a fair comparison, we used a standard supervised baseline, where we did not conduct strong augmentation and the model was not pre-trained with anchor data (as mentioned in the caption of Table 1). Implementing a strong augmentation method is not so common in the benchmark results (such as in FedAdam [1], Fjord [2], and ZeroFL [3]), but for semi-supervised learning, people normally use a strong augmentation method, which we do the same.
> - Description for Figure 3: Please see more explanations in the revised version of the paper, which we highlighted in Section 4.3.
> - Concern about the absence of related work: Thanks for pointing out the papers. We have added FedCon as a baseline (please kindly refer to the general response point number 1 to all reviewers for more details) and included others in the background in our revised manuscript.
> - Regarding other comments: Thanks for the feedback. We have updated our revised manuscript.
>
> [1] Reddi, Sashank, et al. "Adaptive federated optimization." *arXiv preprint arXiv:2003.00295* (2020).
>
> [2] Horvath, Samuel, et al. "Fjord: Fair and accurate federated learning under heterogeneous targets with ordered dropout." *Advances in Neural Information Processing Systems* 34 (2021): 12876-12889.
>
> [3] Qiu, Xinchi, et al. "ZeroFL: Efficient on-device training for federated learning with local sparsity." *arXiv preprint arXiv:2208.02507* (2022).

---

> > ### Comment · Reviewer_6dEL · 2023-11-22
> >
> > 1. The authors have highlighted the label constrastive loss as the main contributions of this paper, however, I am still not convinced the importance of this contribution based on the experiment results.
> >
> > 2.  The supervised baseline: the reason provided by the author does not explain why the supervised learning method has worse performance as the number of anchor data increases and the supervised baseline has the worst performance than other semi-supervised learning method under two special cases (i.e. CIFAR100 when n=10000 and SVHN when n=1000)

---

> ### Author Response · Authors · 2023-11-21
> **Window for responsing and draft updating is closing**
>
> Dear Reviewer #6dEL,
>
> Thanks very much for your time and valuable comments. We understand you're busy. But as the window for responding and paper revision is closing, would you mind checking our response and confirming whether you have any further questions? We are happy to provide answers and revisions to your additional questions. If our reply resolves your concern, please consider raising the score accordingly. Many thanks!
>
> Best regards and thanks,
>
> Authors

---

### Official Review · Reviewer_LCsM · 2023-10-30

**Soundness:** 2 fair
**Presentation:** 1 poor
**Contribution:** 2 fair
**Rating:** 3
**Confidence:** 3

**Summary:**

This work aims at tackling label corruption in FL, called Federated Semi-Supervised Learning, where the server maintains a limited amount of labeled data with unlabelled data on the client side. To this end, the authors provide a double-head structure paired with a contrastive loss. Some experiments are conducted to verify the efficacy of the proposed method.

**Strengths:**

The studied problem is interesting and promising.

**Weaknesses:**

1. The claimed main novelty is the label contraction loss that maximizes the cosine similarity for samples with the same label while minimizing the cosine similarity for samples with different labels. However, this method has been proposed as supervised contrastive learning [1]. Other modules of the proposed method exhibit limited novelty, such as pseudo-labeling and mix-up.

2. The authors seemed to overlook many outstanding works in the literature [2,3]. For instance, the authors involved one method as the baseline method. Moreover, many related works are not considered baseline methods [4].

3. The experimental results are confusing. The proposed method seems to outperform the method with supervised information (See Table 1). This makes the results not convincing.

[1] Supervised Contrastive Learning. Khosla et al. 2020

[2] Federated Semi-Supervised Learning with Inter-Client Consistency & Disjoint Learning. Jeong et al. 2020

[3] FedCon: A Contrastive Framework for Federated Semi-Supervised Learning. Long et al. 2021

[4] pFedKnow, FedIL, and FedSiam

**Questions:**

Suggestions:

1. The paper is hard to follow, I suggest the author improve the writing.
2 Figure 1 provides limited information. For instance, the left figure illustrates the pipeline of FSSL while overlooking the details of the method proposed in this work.

---

> ### Author Response · Authors · 2023-11-17
>
> We sincerely thank the reviewer for their helpful comments and remarks. We now addressed all of the concerns below.
>
> - Concern about novelty: We acknowledge that the general idea of label contrastive loss is not new, but we are the first to leverage it in federated semi-supervised learning to improve the quality of pseudo-labeling. Besides, all existing papers in federated/centralized SSL use the model-predicted results to generate pseudo labels. In our method, we for-the-first-time utilize the latent representation’s cosine similarity for pseudo-labeling and achieve a new SOTA performance. This provides a new solution for the general semi-supervised learning area.
> - Concern about missing literature: Thanks for pointing out the papers. For [2], it is an earlier paper than SemiFL and performs worse than SemiFL, as we mentioned in the background section of our paper. Additionally, we have added FedCon as a baseline in our revised manuscript and included pFedKnow, FedIL, and FedSiam in the background section.
> - Concern about the supervised baseline: For a fair comparison, we used a standard supervised baseline, where we did not conduct strong augmentation and the model was not pre-trained with anchor data (as mentioned in the caption of Table 1). This is the main reason supervised baseline performs poorly in some cases, especially when the anchor set is bigger.

---

> > ### Comment · Reviewer_LCsM · 2023-11-23
> > **Official Comment by Reviewer LCsM**
> >
> > Thanks for your response. However, my concerns remain. Accordingly, I will retain my score as it is.

---

> ### Author Response · Authors · 2023-11-21
> **Window for responsing and draft updating is closing**
>
> Dear Reviewer #LCsM,
>
> Thanks very much for your time and valuable comments. We understand you're busy. But as the window for responding and paper revision is closing, would you mind checking our response and confirming whether you have any further questions? We are happy to provide answers and revisions to your additional questions. If our reply resolves your concern, please consider raising the score accordingly. Many thanks!
>
> Best regards and thanks,
>
> Authors

---

### Official Review · Reviewer_wVdB · 2023-11-01

**Soundness:** 2 fair
**Presentation:** 3 good
**Contribution:** 2 fair
**Rating:** 5
**Confidence:** 3

**Summary:**

The paper introduces "FedAnchor," an innovative Federated Semi-supervised Learning method tailored for the label-at-server scenario. FedAnchor adopts a unique double-head structure, integrating a label contrastive loss based on cosine similarity. This approach optimizes the use of limited labeled anchor data on the server and generates high-quality pseudo-labels, addressing issues like confirmation bias and overfitting common in pseudo-labeling approaches. Besides, the paper conducts extensive experiments across three diverse datasets, demonstrating that FedAnchor outperforms state-of-the-art methods in terms of both convergence rate and model accuracy.

**Strengths:**

1) The proposed label contrastive loss is interesting. Although this new loss is essentially a combination of existing contrastive loss (Info NCE Loss) and the cosine similarity metric, it can be seen from the formula as well as the provided figure that it does bring the same category representations closer and different category representations further apart in the latent feature space. Thus, despite the limited novelty in loss design, I still endorse this as an important contribution.
2) This paper provides a good introduction of the new FSSL method, FedAnchor. In particular, the methodology section has a nice flow and well summarizes the training of FedAnchor proceed on the server and clients at each communication round. The included pseudocode of FedAnchor is easy to understand.
3) FedAnchor addresses an important problem of FSSL research work: the low convergence rate. The experimental results show that FedAnchor significantly prevails the SOTA FSSL baselines in terms of convergence.

**Weaknesses:**

1) The key concern about the paper is the number of SOTA FSSL methods used for comparisons is insufficient. In paper, FedAnchor was compared with supervised learning and SemiFL, while the only meaningful baseline is SemiFL. The finding that a single SOTA FSSL method does not perform as well as fedAnchor is not enough to prove that FedAnchor can outperform all SOTA FSSL methods. Moreover, FedAnchor does not seem to consistently outperform SemiFL. It can be seen in Table 1 that FedAnchor's results on CIFAR100 could be worse than SemiFL when 10000 anchor data was given.
2) Another concern is whether it makes sense in federated learning for a client to download the latent representations of the anchor data from the server to generate pseudo-labels. Federated learning methods often put data privacy at the first priority, so it may be not practical to disclose the complete latent representations of the data on server to each client. Moreover, according to the pseudo-labelling procedure defined in the paper, the obtained latent representations of the anchor data have to be compared with the latent representation of each local data. Under this setting, when the server data and the client's data scale up to a large amount, the effort of pseudo-labelling will be significant. Therefore, I doubt if FedAnchor can be applied to real scenarios.
3) Furthermore, the method section mentions that FedAnchor adopts mixup during the local training on clients. The mixup operation seems to be a crucial trick to improve the accuracy of FedAnchor as shown by the results of Table 1. Why are there no additional ablation studies in the paper on the effect of mixup on FedAnchor convergence and accuracy? Can FedAnchor's performance still be ahead of SemiFL when mixup is not available?

Minor Comment
1) The introduction and background sections are a bit lengthy. Since FedAnchor corresponds to federated semi-supervised learning rather than federated learning, the introduction to federated learning and semi-supervised learning should be shortened.
2) The font sizes of the legends and axis scales in the figures are too small to read.
3) In equation 9, it should be “(x_{fix}, y^{fix})” instead of “(x_{mix}, y^{fix})” if my understanding about the method is not wrong.

**Questions:**

please respond to the weaknesses.

---

> ### Author Response · Authors · 2023-11-17
>
> We sincerely thank the reviewer for their helpful comments and remarks. We now addressed all of the concerns below.
>
> - Concerns about other baseline comparisons. Please kindly refer to the general response to all reviewers.
> - Concerns about the privacy of downloading the latent representations of the anchor data. In our paper, we assume that a small amount of labeled data will be stored on the server, which we call anchor data. We think it is a realistic and practical setup and is mentioned in the introduction. Even if we don’t assume the anchor data is public, since the size of the anchor embedding is significantly smaller than the raw data (we chose 128 in our experiments), it alone is not sufficient to reverse-engineer and reconstruct the raw data.
> - Concerns about scalability. We have addressed this in the general response to all reviewers. Please kindly refer to point number 3 there.
> - Concern about the use of Mixup. Mixup is commonly used in centralized semi-supervised learning settings and is a SemiFL component. We have shown the effect of mixup in our original manuscript (Table 1, Figure 2, 4, 5), named FedAnchor and FedAnchor (mixup).
> - Regarding the font sizes, we will address this for the camera-ready version.
> - Response to Equation 9. Our original equation is correct. Since $x_{mix}$ is already a mixup of $x^{fix}$ and $x^{mix}$, the $y$, as you mentioned, should be as “(x_{mix}, y^{fix})”  for the first part.

---

> ### Author Response · Authors · 2023-11-21
> **Window for responsing and draft updating is closing**
>
> Dear Reviewer #wVdB,
>
> Thanks very much for your time and valuable comments. We understand you're busy. But as the window for responding and paper revision is closing, would you mind checking our response and confirming whether you have any further questions? We are happy to provide answers and revisions to your additional questions. If our reply resolves your concern, please consider raising the score accordingly. Many thanks!
>
> Best regards and thanks,
>
> Authors

---

### Official Review · Reviewer_WkFS · 2023-11-04

**Soundness:** 2 fair
**Presentation:** 3 good
**Contribution:** 2 fair
**Rating:** 3
**Confidence:** 4

**Summary:**

The paper introduces FedAnchor, a novel method for Federated Semi-Supervised Learning (FSSL), which operates under the label-at-server scenario, where only a limited amount of labels are hosted by the server and local clients contain only unlabeled data. This method employs a double-head structure and a label contrastive loss to improve training with limited labeled data at the server and unlabeled client data. FedAnchor aims to reduce issues such as confirmation bias and overfitting, which are common in pseudo-labeling. The method has been tested on three datasets and shows superior performance over current leading methods in both convergence rate and accuracy.

**Strengths:**

1. The considered scenario is novel and practical since the labels from local clients are not trustworthy and can be modeled as missing or noisy.
2. Using anchor points can significantly reduce the confirmation bias induced by pseudo-labeling.

**Weaknesses:**

1. The true contribution of this paper needs to be more clear. Generating pseudo-labels by comparing the similarities between the model representations of unlabeled data and label anchor data (or class prototypes) is not entirely new, e.g., [1--3]. FedSSL is also studied in [4].

2. The paper seems to borrow FixMatch and MixMatch, which is fine. But the contributions in addition to using both algorithm in FL need to be clarified.

3. An important baseline is missing. [4] should be compared in experiments.


[1] Cleannet: Transfer learning for scalable image classifier training with label noise.

[2] Prototypical Pseudo Label Denoising and Target Structure Learning for Domain Adaptive Semantic Segmentation.

[3] Learning from Noisy Data with Robust Representation Learning.

[4] FedCon: A contrastive framework for federated semi-supervised learning.

**Questions:**

Please see Weakness.

---

> ### Author Response · Authors · 2023-11-17
>
> We sincerely thank the reviewer for their helpful comments and remarks. We now addressed all of the concerns below.
>
> - Concern about the novelty: We acknowledge that the general idea of latent representation and cosine similarity is not new in machine learning, which has been used in different areas and applications. The references provided by the reviewer mainly focus on dealing with data/label noise, which is not the concern of our paper. We for-the-first-time leverage the idea of latent representation’s cosine similarity to generate pseudo-labels in semi-supervised learning under the FL setup and achieve a new SOTA performance. The provided references can be good motivation papers, which we have cited in the background section (please see our new manuscript). Also, the method FedCon borrows the idea of using 2 models from BYOL without implementing any contrastive loss, despite the name.
> - Contributions in addition to using FixMatch and MixMatch: 1. A unique pseudo-labeling method in FSSL described above; 2.  A novel label contrastive loss to further improve the quality of pseudo labels.
> - FedCon baseline: We have added this comparison in Table 1, section 4.2. Please kindly refer to general response point number (1) for this point.

---

> > ### Comment · Reviewer_WkFS · 2023-11-21
> >
> > Thank you for the rebuttal and the comparison with FedCon. Could the authors explain:
> > 1. Why does FedCon perform better in Non-IID settings than IID settings?
> > 2. The reported numbers for SemiFL in Table 1 are not consistent with the numbers in SemiFL. Are there any specific reasons? How would the proposed method perform if the experiments were run following the settings in SemiFL?

---

> ### Author Response · Authors · 2023-11-21
>
> We sincerely thank the reviewer for taking the time to review.
>
> - Regarding the first question: If we look at the training curve, we can see that, for this case, the IID curve and the non-IID curve are very similar. We have also attached the plot showing the training curve to Appendix 4 in the newly uploaded manuscript. The testing accuracy between IID and non-IID is similar because, during each round, it also trains on the labeled and balanced anchor data in a supervised learning way. The training using anchor data is supervized and centralized. Also, we observe that solely training on the anchor data with strong augmentation techniques can achieve 40% for anchor size 5000. Through the observation, we can see that the FedCon baseline extracts limited information from the unlabeled data on the client side. Therefore, this reduces the effect of client data distribution (IID/non-IID) for FedCon.
> - Regarding the second question: Since we would like to show that the choice of model architecture would not impact our method, we adopt the more popular and standard model architecture ResNet-18 in our manuscript, but in SemiFL, they use the Wide-ResNet model architecture. Resnet-18 is capable of the image classification task and identifies the "embedding" part of its architecture. Apart from the model architecture, the semiFL implementation, we follows their experimental settings.
>
> We understand you're busy. But as the window for responding and paper revision is closing, would you mind checking our response and confirming whether you have any further questions? We are happy to provide answers and revisions to your additional questions. If our reply resolves your concern, please kindly consider raising the score accordingly. Many thanks!

---

> > ### Comment · Reviewer_WkFS · 2023-11-22
> >
> > Thank you for your explanations. How does the proposed method perform if Wide-ResNet is applied? Changing model architecture without carefully tuning parameters may lead to unfair comparisons.

---

### Author Response · Authors · 2023-11-17
**General response to all reviewers**

We want to take this opportunity to thank all reviewers for their valuable feedback and comments.  We have uploaded the revised version of the paper with the changes highlighted.

We want to make the below few points to all the reviewers:

1. As kindly requested by three reviewers, we have included the FedCon baseline results, as shown in Table 1 in the revised version of the paper. One thing to note is that FedCon borrows the idea of using 2 models from BYOL [1] without implementing any contrastive loss, despite the name. Our implementation of FedCon is based on the original GitHub repository from which we extracted both the client and server training pipeline and put them in our codebase. We replaced the original backbone with ours to perform a fair comparison. We can clearly see from Table 1 that our method outperforms FedCon in all cases.
2. We also conduct ablation studies on the naive implementation of the centralized SOTA method FixMatch under the FL setups, as shown in Table 1 and Section 4.4, to show the effect of the label contrastive loss and the proposed pseudo-labeling. As mentioned in the background section (Sec 2), the original centralized setting of FixMatch requires sampling both labeled and unlabeled data per mini-batch. Therefore, implementing FedAvg+FixMatch combines FixMatch locally on the client side and alternative training using anchor data on the server side. We can see from Table 1 that our method significantly outperforms this baseline.
3. Concerns about scalability: Our method only requires a minor amount of anchor data on the server to obtain acceptable performance. The major overhead in the pseudo-labeling procedure is the computation of n*m cosine similarities, where n is the number of samples on the client and m those on the server. Cosine similarity is well-known for being low complexity and highly optimizable [2]. Such n*m operations are independent and completely parallelizable. Having complete parallelization costs memory because of cloning as many times as necessary. Also, the dimension of such vectors is a constant, 128, which is very small. As a result, it is feasible to scale up to cases where clients and the server contain a large amount of data. We have also mentioned this in the revised version of the paper.

[1] Grill, Jean-Bastien, et al. "Bootstrap your own latent-a new approach to self-supervised learning." *Advances in neural information processing systems* 33 (2020): 21271-21284.

[2]Novotný, Vít. "Implementation notes for the soft cosine measure." *Proceedings of the 27th ACM International Conference on Information and Knowledge Management*. 2018.

---

### Meta-Review · Area_Chair_qLJy · 2023-11-29

**Metareview:**

This paper mainly proposed to use label contrastive loss in federated semi-supervised learning. The major issue is its limited novelty since the final proposal is a combination of existing algorithmic components. Another major issue is its ignorance of several very related papers, which was pointed out by 3 reviewers. As a result, our reviewers consistently agreed to reject it.

**Justification For Why Not Higher Score:**

The major issue is its limited novelty since the final proposal is a combination of existing algorithmic components. Another major issue is its ignorance of several very related papers, which was pointed out by 3 reviewers.

**Justification For Why Not Lower Score:**

N/A

---

### Decision · Program_Chairs · 2024-01-16

Reject